# MTPrompt-PTM: A Multi-Task Method for Post-Translational Modification Prediction Using Prompt Tuning on a Structure-Aware Protein Language Model

**DOI:** 10.3390/biom15060843

**Published:** 2025-06-09

**Authors:** Ye Han, Fei He, Qing Shao, Duolin Wang, Dong Xu

**Affiliations:** 1Department of Electrical Engineering and Computer Science, Christopher S. Bond Life Sciences Center, University of Missouri, Columbia, MO 65211, USA; yhhdb@missouri.edu (Y.H.); hefe@missouri.edu (F.H.); 2Chemical & Materials Engineering, University of Kentucky, Lexington, KY 40506, USA; qshao@uky.edu

**Keywords:** post-translational modification prediction, multi-task prediction, prompt tuning, structure-aware protein language model (S-PLM), knowledge distillation

## Abstract

Post-translational modifications (PTMs) regulate protein function, stability, and interactions, playing essential roles in cellular signaling, localization, and disease mechanisms. Computational approaches enable scalable PTM site prediction; however, traditional models focus only on local sequence features from fragments around potential modification sites, limiting the scope of their predictions. Recently, pre-trained protein language models (PLMs) have improved PTM prediction by leveraging biological knowledge derived from extensive protein databases. However, most PLMs used for PTM site prediction are pre-trained solely on amino acid sequences, limiting their ability to capture the structural context necessary for accurate PTM site prediction. Moreover, these methods typically train separate single-task models for each PTM type, which hinders the sharing of common features and limits potential knowledge transfer across tasks. To overcome these limitations, we introduce MTPrompt-PTM, a multi-task PTM prediction framework developed by applying prompt tuning to a structure-aware protein language model (S-PLM). Instead of training several single-task models, MTPrompt-PTM trains one multi-task model to predict multiple types of PTM sites using shared feature extraction layers and task-specific classification heads. Additionally, we incorporate a knowledge distillation strategy to enhance the efficiency and generalizability of multi-task training. Experimental results demonstrate that MTPrompt-PTM outperforms state-of-the-art PTM prediction tools on 13 types of PTM sites, highlighting the advantages of multi-task learning and structural integration.

## 1. Introduction

Post-translational modifications (PTMs) are crucial regulators of protein function, stability, and interactions. These modifications occur after translation and play essential roles in cellular signaling, protein localization, and disease mechanisms [1,2,3]. Although over 400 distinct PTM types have been identified, most remain poorly characterized regarding their target sites and biological context [4]. Experimental techniques such as mass spectrometry (MS), Western blotting, and radiolabeling are widely used for PTM identification; however, they are expensive, time-consuming, and constrained by technical limitations [5,6,7]. Computational approaches address these challenges by providing fast, cost-effective, and scalable PTM site prediction [8,9,10]. A common practice involves training models on existing PTM datasets to identify potential PTM sites on unseen data. This approach can be broadly categorized into supervised training from scratch or fine-tuning pre-trained protein language models (PLMs).

Training models from scratch typically involves using protein sequence fragments or local structural information through machine learning or deep learning methods. For example, NetPhos 3.1 [10] developed an artificial neural network (ANN) model incorporating sequence-based motifs and structural features to predict phosphorylation sites in eukaryotic proteins. NetNGlyc 1.0 [11] built prediction models for N-linked, O-linked, and C-linked glycosylation sites by utilizing artificial neural networks that examined the sequence context and surface accessibility of potential glycosylation sites. The group-based prediction system (GPS) algorithm in GSP-MSP [12] integrates sequence features to identify specific methylation types on lysine and arginine residues in proteins. MethylSight [13] created a machine learning model that predicts lysine methylation sites in human proteins by utilizing alignment-free features that capture structural information around lysine residues. Ertelt et al. [14] combined machine learning and structure-based protein design to predict and engineer protein post-translational modifications (PTMs), offering a powerful tool for synthetic biology and therapeutic development.

In addition to traditional machine learning techniques, deep learning-based methods have been applied for PTM prediction. For example, MusiteDeep [15] uses convolutional neural networks (CNNs) to automatically learn sequence representations, overcoming the limitations of traditional feature engineering and achieving improved accuracy in phosphorylation site identification. Meanwhile, CapsNet-PTM [16] employs capsule networks to predict seven different PTM types by capturing spatial dependencies between PTM features. Additionally, Wang et al. [17] introduced a web server for the prediction and visualization of 13 PTM types by combining MusiteDeep/CNN and CapsNet deep learning networks and leveraging advanced ensemble techniques. Furthermore, GPS-SUMO 2.0 [18] utilized three advanced machine learning methods—penalized logistic regression (PLR), deep neural networks (DNNs), and Transformer models—to improve the prediction of SUMOylation sites by incorporating multiple sequence features. However, these methods focus only on local sequence features from fragments around potential modification sites, limiting the scope of their predictions.

Training from pre-trained protein language models (PLMs) has proven highly successful in predicting PTM sites. Recently, embeddings from various pre-trained PLMs, such as ESM2 [19], ProtBERT [20], and ProtT5 [20], have been used as features for the training of PTM site prediction models. For instance, Lmnglypred [21] utilizes ProtT5 embeddings to predict N-linked glycosylation sites. PTG-PLM [22] uses embeddings from multiple PLMs, including ProtBERT-BFD, ProtAlbert, ProtXLNet, ESM-1b, and TAPE, to enhance glycosylation and glycation site prediction using CNNs. LM-OGlcNAc-Site [23] applies sophisticated ensemble strategies by combining embeddings from Ankh, ESM-2, and ProtT5 to predict O-linked N-acetylglucosamine (O-GlcNAc) modification sites. In contrast to the aforementioned methods, which rely solely on PLM embeddings as features, PTM-GPT2 [24] fine-tunes a decoder-based autoregressive Transformer model, ProtGPT2, using a custom prompt to guide the model in accurately predicting PTM sites. More recently, PTM-Mamba [25] introduces a novel protein language model that integrates PTM information by incorporating PTM-specific tokens through bidirectional Mamba blocks and fusing them with ESM-2 embeddings using a gating mechanism. The resulting representations can be directly applied to downstream tasks such as phosphorylation and non-histone acetylation site prediction. In contrast to training models from scratch, these methods benefit from embeddings pre-trained on extensive protein sequence databases, allowing them to capture both local sequence motifs (e.g., short patterns around modification sites) and the global sequence context (e.g., long-range dependencies in protein sequences). This makes them highly effective for PTM site prediction. However, these methods have several limitations. First, 3D structural information plays a crucial role in PTM prediction, as most PTMs occur at solvent-exposed residues rather than buried ones, and PTM sites are often influenced by non-local sequence interactions due to protein folding [26]. However, most PLMs used for PTM site prediction are trained solely on amino acid sequences, limiting their ability to capture the structural context necessary for accurate PTM site prediction. Second, different PTM types often occur close to each other on the protein sequence and can share sequence motifs and structural dependencies. However, these methods typically train models for different PTM types separately, preventing the sharing of common features among them. The advantages and disadvantages of the representative PTM prediction tools are presented in Table 1.

To address the limitations mentioned above, we propose MTPrompt-PTM, a novel multi-task PTM site prediction model that leverages the prompt tuning of a structure-aware protein language model (S-PLM) [27] for 13 types of PTM sites, including phosphorylation (S, T, Y), N-linked glycosylation (N), O-linked glycosylation (S, T), ubiquitination (K), acetylation (K), methylation (K, R), SUMOylation (K), succinylation (K), and palmitoylation (C). Our model consists of an encoder and a decoder. The encoder uses S-PLM as the backbone to encode the protein sequence. S-PLM is a pre-trained PLM incorporating structural information with sequence-based embeddings from ESM2. During the multi-task training phase, all parameters of S-PLM are frozen. However, to effectively leverage the pre-trained model’s information, we perform prompt tuning on S-PLM. Prompt tuning [28] is a parameter-efficient fine-tuning (PEFT) technique that adds trainable embeddings, called ‘prompts’, to the sequence embeddings. Unlike full fine-tuning, where all model weights are updated, prompt tuning optimizes only the additional trainable embeddings, reducing the computational overhead while maintaining generalization. In our model, we propose a novel method for the initialization of our task prompts. The decoder consists of shared layers and task-specific layers, which capture common features and task-specific features separately. Additionally, we incorporate a knowledge distillation strategy, where single-task models teach a multi-task model, helping the multi-task model to outperform its single-task counterparts by integrating knowledge across multiple tasks. Our experimental results show that MTPrompt-PTM improves the predictive performance compared to single-task models. To further validate its effectiveness, we compare MTPrompt-PTM with state-of-the-art PTM prediction tools. The results demonstrate that MTPrompt-PTM outperforms these tools across all 13 PTM types, confirming its effectiveness.

## 2. Materials and Methods

### 2.1. Dataset and Data Processing

Numerous databases and research studies provide PTM data; however, most of them offer only short peptide fragments centered around the modified residue, lacking a full protein context or complete sequence information. These truncated sequences often miss the essential global context, making it difficult to capture long-range interactions between residues, which can be critical in determining PTM occurrence. To address this limitation, we utilize full-length protein sequences in our study, enabling the model to capture the comprehensive contextual information and long-range sequence dependencies necessary for accurate PTM site prediction.

UniProt, the largest protein sequence database with PTM annotations, contains over 200 million protein sequences and provides annotations for 200 PTM types. Therefore, we constructed a new PTM dataset from UniProt, incorporating 13 PTM types: phosphorylation (S, T, Y), N-linked glycosylation (N), O-linked glycosylation (S, T), ubiquitination (K), acetylation (K), methylation (K, R), SUMOylation (K), succinylation (K), and palmitoylation (C).

To build this dataset, we first downloaded full-length protein sequences along with their PTM annotations for the 13 PTM types from UniProt. The data were then filtered by species and sequence length, retaining only metazoan proteins and excluding sequences longer than 1022 residues. Sequences longer than 1022 were not processed because longer sequences would have exceeded the model’s input size limit and negatively affected the Transformer model’s performance. Processing excessively long sequences can lead to memory overload, slower processing times, and reduced effectiveness due to the quadratic complexity (O(n^2^)) of the attention mechanism. We chose not to truncate the sequences because truncation could result in losing important functional or structural information, especially for long protein sequences, where key motifs or functional sites might be outside the truncated region. By limiting the sequence length to 1022 tokens, we ensured that we retained the most relevant parts of the sequence without losing critical details, striking a balance between computational efficiency and maintaining the essential sequence context. Table 2 presents the PTM types, the corresponding UniProt annotations, and the number of protein sequences.

For each protein sequence, the PTM sites are treated as positive samples, while other positions with the same amino acids, excluding the PTM sites, are treated as negative samples. Figure 1 shows the number of PTM sites in terms of positive and negative sites for each PTM type. During training, the entire protein sequence is input, but the loss is calculated only for the positive and negative sites.

We then separated all the protein sequences into a training set and a testing set based on the timestamp. Protein sequences annotated in UniProt prior to 2010 were used for training, while those annotated after 2010 were reserved for testing. We trained our model on the training data and used the test data to compare our model’s performance with that of other state-of-the-art tools. Additionally, we applied the widely used clustering program CD-HIT-2D to assess the similarity between the training and testing data. The testing protein sequences with no more than 60%, 70%, and 80% similarity to the training data were generated using CD-HIT-2D. We present the performance of the testing data at different levels of sequence similarity to the training data.

Furthermore, we created another non-redundant dataset to evaluate our model. We applied CD-HIT [29] to cluster this dataset based on a 60% sequence similarity threshold. To avoid homologous redundancy, only one representative sequence from each cluster was selected. The non-redundant dataset was then split into training and testing sets in a 4:1 ratio. These datasets were used to train and evaluate our model, ensuring a robust performance assessment.

We further evaluated our model using an independent benchmark for phosphorylation and non-histone acetylation. The phosphorylation test set was obtained from the ProteinBERT benchmark [30], which is derived from PhosphoSitePlus [31], a comprehensive resource of experimentally validated post-translational modifications in human and mouse proteins. The non-histone acetylation test set was sourced from TransPTM [32], a Transformer-based model specifically designed for the prediction of non-histone acetylation sites.

### 2.2. Architecture of MTPrompt-PTM

This paper introduces MTPrompt-PTM, a multi-task model for post-translational modification prediction. The overall architecture of MTPrompt-PTM is illustrated in Figure 2. Our model includes an encoder and decoder. The encoder leverages S-PLM v2 as its backbone and is trained using prompt tuning with task prompts. The decoder is a hybrid architecture comprising shared feature extraction layers and task-specific classification layers.

Unlike most PTM prediction methods that use peptides as input, our model takes entire protein sequences as input. Initially, the protein sequences containing PTM sites are tokenized using the ESM2 tokenizer, which converts them into sequence embeddings. Task prompts, which act as additional trainable embeddings to guide the model in distinguishing between different PTM prediction tasks, are concatenated with the protein sequence embedding. This combined matrix is then passed through the encoder as usual. Throughout the multiple Transformer layers in S-PLM v2, the model generates updated task prompts and protein sequence embeddings. After the Transformer layers, the task prompts, along with the [CLS] and [EOS] tokens, are discarded. [CLS] (Classification) and [EOS] (End of Sequence) are special tokens commonly used in Transformer-based language models. The [CLS] token is typically added at the beginning of an input sequence, and its corresponding output embedding is used for classification tasks, summarizing the entire input. The [EOS] token marks the end of a sequence, signaling the model where the input terminates, which is particularly important in generative or sequential prediction tasks. Only the residue-level embeddings for the sequence are retained and passed to the decoder for further processing. During the entire training process, while the parameters of S-PLM remain frozen, the task embeddings are updated through gradient descent.

Our decoder is designed with both shared layers and task-specific layers. The shared layers consist of two CNN Inception modules and a fully connected (FC) layer. Each CNN Inception module is composed of three 1D CNN layers with different filter sizes, enabling the capture of multi-dimensional local information from the input sequences. The outputs from these two modules are concatenated to combine the captured local features. Following the CNN layers, a fully connected layer is added to further capture and refine the information. These shared layers are responsible for learning common, generalizable representations that can be applied across different post-translational modification (PTM) types. Once the shared representation is learned, task-specific classification layers are introduced to handle the unique characteristics of each PTM type. These task-specific layers consist of 13 fully connected layers, each corresponding to a different PTM type. These layers can be seen as 13 independent classification heads, with each head trained to focus on the specific sequence patterns, structural features, or biochemical properties associated with its respective PTM. Each PTM-specific head independently predicts the probability of the presence or absence of its respective modification at each relevant sequence position. By maintaining separate classification heads for each PTM type, the model ensures that features are tailored for each modification, enhancing the model’s predictive accuracy and allowing for more precise modeling of the diverse PTM signals.

### 2.3. Prompt Tuning on MTPrompt-PTM

In our encoder, to generate residue-level embeddings with enhanced structural information, we use S-PLM v2 [33] as the backbone. S-PLM [27] is a structure-aware protein language model integrating both sequence and structural information via contrastive learning. S-PLM v2 is the upgraded version of S-PLM, using a geometric vector perceptron (GVP) model [34] to achieve more precise residue-level embeddings by capturing detailed geometric properties. The sequence encoder of S-PLM builds upon a pre-trained ESM2 model, preserving previously learned protein knowledge while effectively adapting to new tasks. In contrast to ESM2, S-PLM explicitly incorporates structural information due to its pre-training on paired protein sequences and contact maps, enabling the direct encoding of spatial relationships and residue–residue interactions into its representations. T-SNE clustering results have shown that S-PLM achieves superior kinase group clustering compared to ESM2, underscoring its potential as an effective backbone model for PTM site prediction [27].

Although S-PLM already contains rich general knowledge learned from large-scale protein sequence data, we use prompt tuning to make task-specific adjustments for the prediction of post-translational modifications. The core idea of prompt tuning is to concatenate the task prompts with the protein sequence embeddings and input them into the pre-trained language model. This allows the Transformer operations to be performed while keeping the original model’s weights frozen. As a result, the final protein sequence embeddings are adjusted by the task prompts. Given that prompt tuning is highly sensitive to the initialization of the task prompts, it is crucial to initialize the prompts effectively. Therefore, we propose a novel initialization method for different tasks. First, we collect all the protein sequences from the training set and obtain their sequence embeddings by inputting them into S-PLM v2. Next, we extract 21-residue peptides centered on the PTM site and compute the average of their embeddings. These averages are then clustered into K clusters. Finally, we average the values within each cluster to generate the final prompt matrix. The specific initialization process is outlined below.

Step 1. Extracting Protein Embeddings

We utilized S-PLM v2 to generate embeddings for every residue in the training protein sequences. By feeding the entire training set into S-PLM, we obtained residue-level embeddings with a shape of *N* × 1280, where *N* represents the total number of residues. The value 1280 corresponds to the dimensionality of the embedding vector produced by S-PLM v2 for each residue. These embeddings capture rich, context-aware biochemical and structural information for each amino acid, providing a robust foundation for downstream PTM prediction.

Step 2. Generating PTM-Centered Embeddings

Since PTMs are often influenced by the local sequence environment surrounding the modified residue, we extract a 21-residue window centered on each PTM site to capture this context. This window includes a modified residue along with its ten upstream and ten downstream neighbors, effectively preserving the immediate biochemical environment. For each PTM site, the contextual window S_i_ is defined as(1)$$S_{i}=\left\{E_{i-10},E_{i-9},\ldots ,E_{i},\ldots ,E_{i+9}E_{i-10}\right\},\;S_{i}\in R^{21\times 1280}$$
where $E_{i}$ represents the embedding of the *i*-th residue. These windows provide a localized, high-dimensional representation of the sequence, which is essential for accurate PTM site modeling.

Step 3. Computing Mean Embeddings

Each window is then averaged to produce a single 1280-dimensional vector that represents the local environment of the PTM site. The mean embedding *E_i_* for each PTM site is computed as(2)$$E_{i}=\frac{1}{21}\sum_{j=i-10}^{i+10}E_{j},\;E_{i}\in R^{1280}$$

This mean pooling simplifies the representation while retaining the essential pattern of this PTM context.

Step 4. Clustering PTM Representations

Instead of averaging all PTM site embeddings into a single global representation, we apply K-means clustering to group them into K distinct clusters. After experimenting with different values of *K*, we found that the best results were achieved when *K* = 500. Each cluster captures a recurring motif or feature pattern shared across different proteins, preserving the diversity and subtlety of PTM-specific contexts. Formally, each cluster *C_k_* is defined as(3)$$C_{k}=\left\{E_{i}|i\in cluster\;k\right\}$$

The rationale behind selecting *K* clusters is to preserve the inherent diversity and fine-grained patterns captured in the embedding space. Biological modifications (PTMs) often exhibit subtle yet meaningful variations, reflecting different functional contexts, regulatory mechanisms, or kinase substrate specificity. By clustering the embeddings into multiple distinct groups, we maintain these biologically relevant variations, allowing each cluster to represent a unique pattern or functional state more accurately. Direct averaging could obscure these subtle differences and lead to the loss of critical biological insights.

Step 5. Computing Cluster Centroids

For the *k*-th cluster, the centroid vector (the mean of the embeddings in this cluster) $\mu_{k}$ is computed as(4)$$\mu_{k}=\frac{1}{\left|C_{k}\right|}\sum_{E_{i}\in C_{k}}E_{i},{\;\;\;\mu }^{k}\epsilon R^{1280}$$

These centroids serve as prototypical representations of common PTM-related contexts.

Step 6. Constructing the Final Prompt Matrix

The 500 centroids obtained from clustering are then stacked to form a matrix that serves as a task-specific embedding. This matrix acts as a learnable guide for the model, capturing diverse PTM-related patterns and functional contexts. The final prompt embedding matrix *M* has a shape of *K* × 1280 and is defined as(5)$$M=[\mu_{1};\mu_{2};\ldots ;\mu_{K}]\epsilon R^{K\times 1280}$$

### 2.4. Multi-Task Training of MTPrompt-PTM

To improve both the performance and generalizability of the multi-task model, we adopt a knowledge distillation strategy, which transfers knowledge from a teacher model to a student model by training the student to imitate the teacher’s outputs. Unlike traditional training with one-hot labels, the teacher’s probability distribution over classes, referred to as soft labels, provides a richer and more informative learning signal. Clark et al. [35] demonstrated that using a single-task teacher model to guide a multi-task student model is significantly more effective than employing multiple teachers for multiple tasks. This is because the student benefits from exposure to a diverse set of PTM-specific teachers, similarly to how ensemble learning enhances generalization. Inspired by this, we apply knowledge distillation in our framework by using single-task models as teachers to train the multi-task model, enabling it to leverage both expert knowledge and shared task representations for improved performance.

As shown in Figure 3, the training process consists of two main steps. In Step 1, we independently train 13 single-task models for all 13 PTM types. These single-task models act as teacher models, with architectures nearly identical to that of the multi-task model, except for the absence of task-specific layers. After training, we use the single-task models to generate predictions for all training data. These predictions, serving as soft labels, capture subtle patterns and uncertainties often missed by traditional hard labels. In Step 2, we merge the training data from all PTM types to train the multi-task student model. The soft labels from the teacher models are used to guide the student model’s training. To further enhance the training, we adopt a teacher annealing strategy [35], progressively blending the teacher’s soft labels with ground truth annotations. This combination provides a refined supervisory signal, improving the model’s generalization and accuracy across diverse PTM types. To address potential class imbalances from simply concatenating all datasets, we apply a weighted loss function that combines soft labels (from teacher models) and hard labels (ground truth annotations). The weights are determined based on the dataset size, and the loss function is defined as(6)$$L(\theta )=\sum_{\;t\epsilon T}{weight}_{t}\sum_{x_{t}^{i}y_{t}^{i}\epsilon D_{t}}l(\gamma y_{t}^{i}+\left(1-\gamma \right)f_{t}\left(x_{t}^{i},\theta_{t}\right),f_{t}\left(x_{t}^{i},\theta \right)),\;\gamma \in (0,1)$$

Here, T denotes the set of PTM tasks. For each task t, we first train a single-task teacher model with parameters $\theta_{t}$ and then use its predictions to guide the multi-task student model with parameters θ. The variable γ controls the balance between hard and soft labels during training; specifically, we set γ = 0.5 to dynamically reconstruct the training labels by averaging the teacher’s soft predictions and the ground truth.

## 3. Results

### 3.1. Comparison with State-of-the-Art Tools

Here, we compare MTPrompt-PTM with several state-of-the-art PTM prediction tools, including MusiteDeep [17], PTMGPT2 [24], NetPhos3.1 [10], NetOGlyc4.0 [36], NetNGlyc1.0 [11], GPS-SUMO2.0 [18], CSS-Palm4.0 [37,38], GSP-MSP [12], and MethylSight [13]. Table 2 presents the prediction results on the test set for all 13 PTM types, including phosphorylation (S, T, Y), N-linked glycosylation (N), O-linked glycosylation (S, T), ubiquitination (K), acetylation (K), methylation (K, R), SUMOylation (K), succinylation (K), and palmitoylation (C).

As shown in Table 3, MTPrompt-PTM outperformed all other tools across all 13 PTM types. For example, in terms of the Matthews correlation coefficient (MCC), phosphorylation (S) exhibited a substantial improvement, with MTPrompt-PTM achieving a 118.9% increase in the MCC compared to MusiteDeep. This trend continued with phosphorylation (T, Y), where our model outperformed MusiteDeep by 58.0% and 26.5%, respectively. In the case of O-linked glycosylation (S, T), MTPrompt-PTM showed improvements of 24.2% and 63.6% over MusiteDeep. For N-linked glycosylation (N), MTPrompt-PTM showed a smaller improvement of 3.2%. For SUMOylation (K), MTPrompt-PTM outperformed PTMGPT2 by 46.2%, while MusiteDeep did not achieve comparable results. This difference can be attributed to the fact that MusiteDeep uses a smaller dataset than the one used to train our model. Similarly, for ubiquitination (K) and acetylation (K), our model exceeded MusiteDeep by 140.3% and 16.7%, respectively. Since MusiteDeep does not provide a model for the prediction of succinylation, we compared MTPrompt-PTM only with PTMGPT2 for this PTM type. Here, our model achieved a 104.2% improvement over PTMGPT2 in succinylation (K). For palmitoylation (C), methylation (R), and methylation (K), MTPrompt-PTM surpassed MusiteDeep by 28.5%, 50.7%, and 13.3%, respectively. These results demonstrate that our model outperforms many existing PTM prediction tools and can be considered a leading tool for PTM site prediction.

Figure 4 and Figure 5 present the ROC curves and precision–recall curves on the test set for all PTM types across different methods. Since PTM-GPT2 only provides binary predictions (i.e., whether a residue is a PTM site or not) without probability scores, we could not plot its full ROC and precision–recall curves. However, we calculated its TPR, FPR, precision, and recall to mark PTM-GPT2 as a single point on the curves. The AUC and PRAUC values of all methods are consistent with the results in Table 2, further demonstrating that our model outperforms other approaches.

To further evaluate the kinase-level specificity and robustness of MTPrompt-PTM, we conducted a comparative analysis across several kinase families. Table 4 summarizes the number of kinase sites included in both the training and testing sets, as well as the number of correctly predicted kinase sites and the corresponding accuracy for each model. Our results demonstrate that MTPrompt-PTM consistently achieves high accuracy across most kinase families. Particularly in the CMGC and AGC families, which have relatively large numbers of kinase sites in both the training and testing sets, MTPrompt-PTM significantly outperforms other models, indicating its strong generalization abilities in data-rich settings. These findings suggest that our multi-task prompt tuning framework and structure-aware backbone not only improve overall PTM prediction but also enhance kinase-specific site recognition. This highlights the potential of MTPrompt-PTM as a generalizable tool for kinase-centered PTM analysis. Notably, all models perform poorly on the “other” kinase family, with MTPrompt-PTM achieving accuracy of only 0.111 and the remaining models showing similarly low or inconsistent performance. This may be attributed to the relatively limited number of training samples available for this group (only 415 sites, compared to over 1700 for CMGC or 879 for AGC), which constrains the model’s ability to learn meaningful and generalizable features for these kinases. Additionally, the “other” category likely includes a diverse and heterogeneous set of kinases that do not share common sequence or structural motifs, further complicating the prediction task. This observation highlights a common challenge in PTM prediction: models tend to favor well-represented kinase families during training (e.g., AGC and CMGC), and their performance may degrade when applied to underrepresented or diverse groups (e.g., other).

Figure 6 illustrates the performance on test subsets with varying levels of sequence similarity to the training data, evaluated using the F1 score. The test subsets, containing protein sequences with no more than 60%, 70%, 80%, 90%, and 100% similarity to the training set, were generated using CD-HIT-2D. Across nearly all PTM types, MTPrompt-PTM consistently outperformed other methods regardless of the sequence similarity thresholds. However, for certain PTMs, we observe a noticeable decline in MTPrompt-PTM’s performance as the sequence similarity decreases. This degradation can be attributed to several factors. Some PTM types may have fewer annotated examples or less diversity in the training set. As a result, the model may be overfit to specific sequence patterns and struggle to generalize to more dissimilar sequences. In addition, certain PTMs, such as phosphorylation (S) and SUMOylation (K), are known to be highly context-dependent, often influenced by local sequence motifs or secondary structure elements. When the sequence similarity drops, these subtle cues may no longer be preserved, making it more challenging for the model to accurately identify modification sites. Another contributing factor may be the imbalance in positive and negative samples across the different similarity subsets. As the similarity threshold decreases, the number of true PTM sites that remain in the test set may decline disproportionately, leading to a more severe class imbalance and potentially skewing the model’s predictions. Despite these challenges, MTPrompt-PTM still maintains relatively strong performance across all similarity levels, underscoring its robustness compared to existing tools.

Figure 7 compares the performance of MTPrompt-PTM with that of MusiteDeep, PTMGPT2, and PTM-Mamba on independent benchmark datasets for phosphorylation and non-histone acetylation. For phosphorylation, MTPrompt-PTM consistently achieves the best overall performance, particularly excelling in its accuracy, precision, F1 score, and MCC. This suggests that our model makes more reliable and balanced predictions with fewer false positives and the better discrimination of true modification sites. PTM-Mamba shows the highest recall, indicating its sensitivity in detecting true sites, but this comes at the cost of lower precision, leading to more false positives. In non-histone acetylation, MTPrompt-PTM again outperforms other methods in terms of accuracy and precision, demonstrating its ability to correctly identify modification sites with fewer errors. While MusiteDeep and PTM-Mamba have comparable recall values, their lower precision reduces their overall predictive quality. The differences may be due to MTPrompt-PTM’s multi-task prompt tuning strategy and structure-aware embedding, which enhance its specificity and robustness across diverse PTM types. Overall, these results highlight that MTPrompt-PTM strikes a better balance between sensitivity and specificity, making it a more effective tool for PTM site prediction compared to competing methods.

### 3.2. Comparison with Single-Task Models on Different PTM Types

To evaluate the effectiveness of our proposed multi-task architecture, we compared it against 13 independently trained single-task models on our non-redundant dataset, with each model trained on a single PTM type. Both the multi-task and single-task models shared the same pre-trained backbone and utilized the same predefined task tokens, ensuring a fair comparison.

The results in Figure 8 indicate that phosphorylation (S, T, and Y) exhibited varying degrees of improvement in the multi-task model. Phosphorylation (S) showed only a 2% improvement in the AUPRC, likely due to its large training dataset, which may reduce its dependency on knowledge transfer from other PTM types. In contrast, phosphorylation (T) showed substantial gains, with threonine improving by 9% (AUPRC), suggesting that phosphorylation (T) benefits from shared knowledge with phosphorylation (S). O-linked glycosylation (S and T) demonstrated strong improvements across all metrics, with O-linked glycosylation (S) achieving improvements of 11.1% (AUPRC) and O-linked glycosylation (T) seeing the highest gains, with the AUPRC increasing by 11.2%, suggesting a possible interaction with phosphorylation. N-linked glycosylation (N) performed slightly worse in the multi-task setting, with an AUPRC decrease of 0.1%, indicating that asparagine may be more independent and less influenced by other PTM types. Acetylation (K), ubiquitination (K), succinylation (K), and SUMOylation (K) benefited from the multi-task model, showing improvements of 9.5%, 2.9%, 10.2%, and 2.5% (AUPRC), suggesting that PTMs occurring on lysine (K) may support each other through shared information. Overall, the results demonstrate that multi-task learning improves the performance for PTMs that share functional or structural similarities, such as phosphorylation and O-linked glycosylation. However, PTMs that are more independent, such as N-linked glycosylation (N), may not benefit as much from multi-task training. Additionally, PTMs on lysine (K) appear to influence each other, as seen in the gains for acetylation, ubiquitination, SUMOylation, and succinylation. The ROC curves are shown in Appendix A. From the AUROC, we observe that the performance of the multi-task model is similar to that of the single-task model. This could be due to the imbalance between negative and positive samples. The improved AUPRC of the multi-task model likely arises from its ability to generalize across multiple PTM types, which helps to enhance its precision and recall for the positive class, especially in the presence of class imbalances. While the multi-task setup does not significantly impact the AUROC (as the AUROC is less sensitive to class imbalances), it has a more pronounced effect on the precision–recall performance.

The F1 and MCC can show the improvements in the multi-task model as well, as presented in Table 5. From this table, we can see that MTPrompt-PTM achieves better performance than the single-task model in most PTM types. Notably, for phosphorylation (T), phosphorylation (Y), O-linked glycosylation (S and T), and SUMOylation (K), the multi-task model yields higher F1 and MCC scores, indicating that leveraging shared knowledge across tasks enhances the prediction accuracy for these modification types. For example, in O-linked glycosylation (T), the F1 and MCC scores of the multi-task model reach 0.716 and 0.685, respectively, outperforming the single-task model (0.670/0.634). Although the single-task model performs slightly better in a few PTMs, such as N-linked glycosylation (N) and methylation (R), the difference is marginal, suggesting that the multi-task framework does not significantly compromise the performance even in well-characterized or distinct PTMs. Overall, the results demonstrate that the multi-task model provides more stable and generalized performance across diverse PTM types, especially those with limited training data or weaker individual signals.

### 3.3. Ablation Study

#### 3.3.1. Comparison with Multi-Task Model Without Knowledge Distillation on Different PTM Types

To assess the impact of knowledge distillation, we compared MTPrompt-PTM with a baseline multi-task model trained solely on hard labels without distillation. As shown in Table 6, MTPrompt-PTM consistently achieved higher F1 and MCC scores across all PTM types, demonstrating improved predictive accuracy and robustness.

Table 7 presents the AUROC and AUPRC comparisons, where MTPrompt-PTM achieves notably higher AUPRC values in most cases. For instance, in O-linked glycosylation (S) and (T), the AUPRC increased from 0.518 and 0.761 to 0.552 and 0.784, respectively. This improvement in the AUPRC is particularly significant given the class imbalance typically present in PTM site prediction tasks.

These results confirm that incorporating soft labels from single-task models during training enables the multi-task framework to capture richer probabilistic information and subtle interdependencies across PTM types. This additional knowledge improves its generalization, especially for low-signal or data-scarce PTMs, ultimately leading to better overall performance than training on hard labels alone.

#### 3.3.2. Comparison with Fine-Tuning the Last Two Layers of S-PLM on Different PTM Types

To further evaluate the effectiveness of our prompt tuning strategy, we compared MTPrompt-PTM with a commonly used fine-tuning approach that updates only the last two layers of the pre-trained S-PLM model. This comparison assessed whether prompt tuning could match or exceed the performance while minimizing the computational overhead. As shown in Table 8, MTPrompt-PTM achieved higher F1 and MCC scores across nearly all PTM types. Notably, large gains are observed in phosphorylation (T) and O-linked glycosylation (T), where the F1/MCC scores increase from 0.403/0.392 and 0.548/0.554 to 0.461/0.432 and 0.716/0.685, respectively. These results indicate that prompt tuning significantly enhances both the precision and robustness in PTM site prediction. Although the fine-tuned model slightly outperforms MTPrompt-PTM in ubiquitination (K), methylation (K), and methylation (R), the performance gap is minimal.

Table 9 further confirms this trend through AUROC and AUPRC comparisons. MTPrompt-PTM shows a notable advantage in the AUPRC, especially for PTMs like O-linked glycosylation (S) and O-linked glycosylation (T), with improvements from 0.525 and 0.757 to 0.552 and 0.784, respectively. These metrics are particularly important in highly imbalanced datasets, where the ability to correctly identify true positives is critical. Furthermore, in phosphorylation (Y), the AUPRC improves from 0.499 to 0.504, and, in SUMOylation (K), from 0.354 to 0.369, reinforcing the consistency of the performance gains across diverse PTM types.

This improvement may be attributed to two key factors. First, by keeping the entire ESM2 model frozen, MTPrompt-PTM retains the broad, general-purpose protein representations learned during large-scale pre-training, avoiding the risk of overfitting or forgetting. Second, the integration of task-specific prompt embeddings enables fine-grained adaptation to each PTM type, capturing subtle biochemical cues that are often lost in shallow fine-tuning. In contrast, updating only the last two layers may be insufficient to extract the deep contextual information required for accurate PTM site identification.

## 4. Discussion

Exposing a model to a diverse set of tasks can serve as an effective form of regularization, reducing the risk of overfitting by encouraging the learning of generalizable patterns rather than memorizing task-specific details. A key advantage of multi-task learning lies in its ability to facilitate knowledge transfer between related tasks, i.e., improvements in one task can enhance the performance in others. Building on this principle, we developed MTPrompt-PTM, the first multi-task PTM prediction model capable of predicting 13 types of post-translational modifications (PTMs): phosphorylation (S, T, Y), N-linked glycosylation (N), O-linked glycosylation (S, T), ubiquitination (K), acetylation (K), methylation (K, R), SUMOylation (K), succinylation (K), and palmitoylation (C). At inference time, users simply need to provide a protein sequence along with the PTM type(s) that they wish to predict, making MTPrompt-PTM both versatile and user-friendly.

Unlike conventional PLM-based methods, MTPrompt-PTM leverages multi-task prompt tuning on the pre-trained S-PLM model, allowing it to adapt to diverse PTM types by incorporating task-specific signals. A decoder architecture composed of shared and task-specific layers further enables the model to capture both general and PTM-specific representations during training. To enhance the performance and generalization, knowledge distillation is employed, transferring insights from multiple single-task teacher models into a unified multi-task student model. Through extensive comparisons with single-task models and several state-of-the-art PTM prediction tools, MTPrompt-PTM consistently outperforms alternative methods across all PTM types, affirming the effectiveness of multi-task learning within this domain.

The effectiveness of MTPrompt-PTM can be attributed to three key factors. First, MTPrompt-PTM leverages the S-PLM v2 backbone, which captures both local and global sequence and structural information, providing a strong foundation for PTM prediction. Second, it employs multi-task prompt tuning, a lightweight fine-tuning method that efficiently adapts the PLM to the nuances of multiple PTM types while retaining the general-purpose knowledge encoded in the protein language model. This approach enables PTM prediction without compromising the pre-trained model’s integrity. Third, MTPrompt-PTM incorporates a multi-PTM training framework with a knowledge distillation strategy, facilitating shared learning across different PTM types. This strategy enhances the performance, particularly for PTMs with limited training data.

However, several limitations exist. First, due to the computational complexity of processing long sequences, MTPrompt-PTM can only accept protein sequences of up to 1022 residues. This limitation could restrict its applicability in real-world scenarios, where longer sequences are common. Second, to better simulate real-world conditions, we separated the training and testing sets based on timestamps. However, some similarities between the training and testing sets remained. As the sequence similarity between the training and testing sets decreases, the performance also decreases. This could be due to data leakage, where information from the testing set unintentionally influences the training process, potentially causing overfitting. As a result, the model becomes overly specialized in sequences present in the training set and struggles to generalize to novel sequences in the test set.

In the future, we aim to extend our framework to support continuous learning, enabling it to accommodate additional modifications as new data become available. Expanding the dataset and incorporating more diverse annotations will improve the generalizability. However, challenges related to ensuring that continuous learning does not interfere with the performance of previous models need to be addressed. Additionally, the imbalanced nature of PTM training data may lead to biased predictions toward overrepresented classes. Future work should explore techniques such as focal loss, class reweighting, or data augmentation to address this imbalance and improve the model’s fairness and accuracy.

## 5. Conclusions

MTPrompt-PTM represents a step forward in the scalable, multi-task prediction of post-translational modification sites. Instead of relying on fragmented single-task approaches, it unifies 13 PTM types into one flexible framework, enabling users to make efficient, type-specific predictions from a single model. Its consistent performance across benchmark datasets, kinase-specific analyses, and external validation scenarios underscores its potential for broad application in bioinformatics. Looking ahead, MTPrompt-PTM provides a foundation for continuous, modular learning as PTM databases expand, supporting future developments in multi-label PTM annotation at the proteome scale.

## Figures and Tables

**Figure 1 biomolecules-15-00843-f001:**
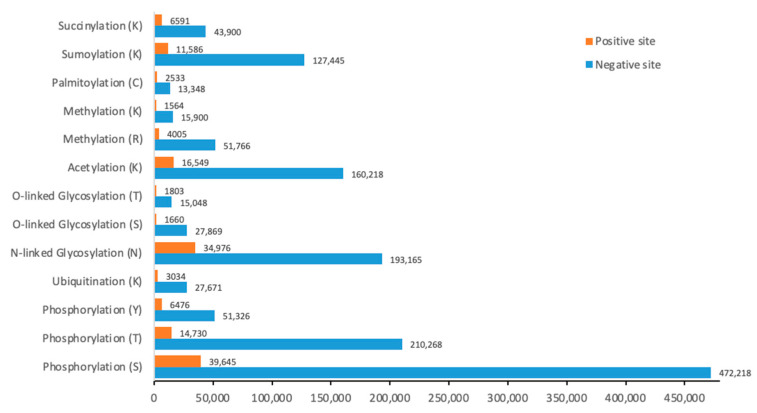
The distribution of positive and negative PTM sites across different PTM types.

**Figure 2 biomolecules-15-00843-f002:**
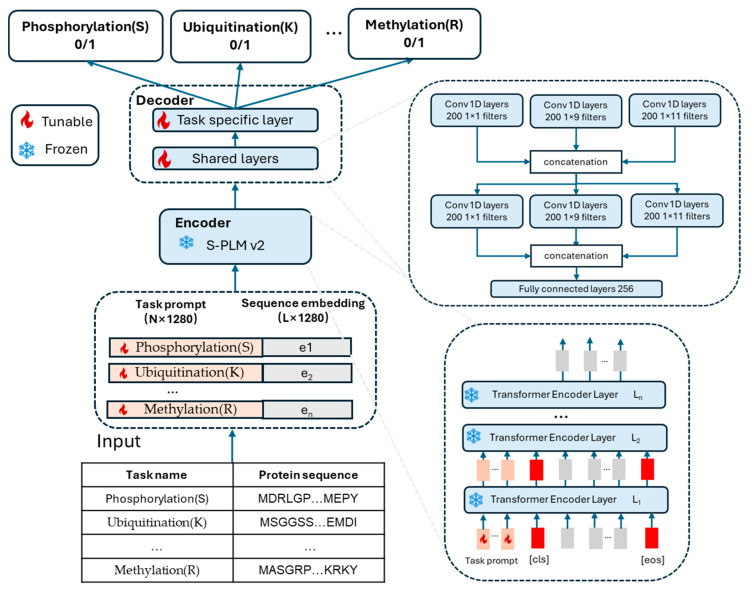
**The architecture of MTPrompt-PTM**. During multi-task training, the model takes the task name and protein sequence as input. The task prompts are initialized using our proposed method based on the task name. The protein sequence is tokenized and embedded using the ESM2 tokenizer. The task prompts are then concatenated with the sequence embeddings and input into the encoder. The backbone of the encoder is S-PLM v2, with the input passing through 33 Transformer encoder layers. Then, residue-level representations (excluding [CLS], [EOS], and task tokens) are extracted and passed to the decoder. During the entire training process, while the parameters of S-PLM remain frozen, the task prompts are updated through gradient descent. The decoder features a hybrid architecture with shared and task-specific layers. The shared component consists of two CNN Inception modules, each containing three 1D convolutional layers with varying kernel sizes, followed by concatenation and a fully connected layer. The task-specific layers process the shared residue representation and perform classification. Each task-specific head corresponds to a different PTM type, receiving residue representations and outputting whether the residue belongs to the respective PTM type.

**Figure 3 biomolecules-15-00843-f003:**
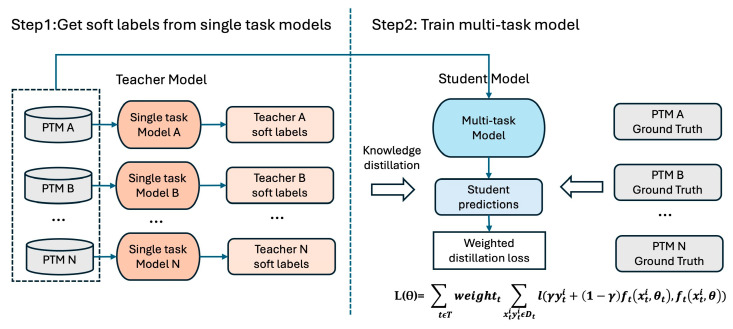
**Training process of MTPrompt-PTM**. In Step 1, we independently train 13 single-task models, each corresponding to a different PTM type, which serve as teacher models. The saturated orange ovals are teacher models. These models have architectures similar to that of the multi-task model, with the key difference being the absence of task-specific layers. The teacher models generate soft labels through predictions on the training data, capturing subtle patterns and uncertainties missed by traditional hard labels. The paler peach boxes immediately to their right are the soft-label outputs those teachers produce. In Step 2, we merge the training data from all PTM types to train the multi-task student model. The powder-blue oval is the student model that learns all tasks jointly. The soft labels from the teacher models guide the student model’s training. A teacher annealing strategy is applied, progressively blending the teacher’s soft labels with ground truth annotations to improve the model’s generalization and accuracy. The light-blue rectangle below it is the student’s prediction. To address class imbalances, we use a weighted loss function that combines both soft labels and hard labels, with weights determined by the dataset size.

**Figure 4 biomolecules-15-00843-f004:**
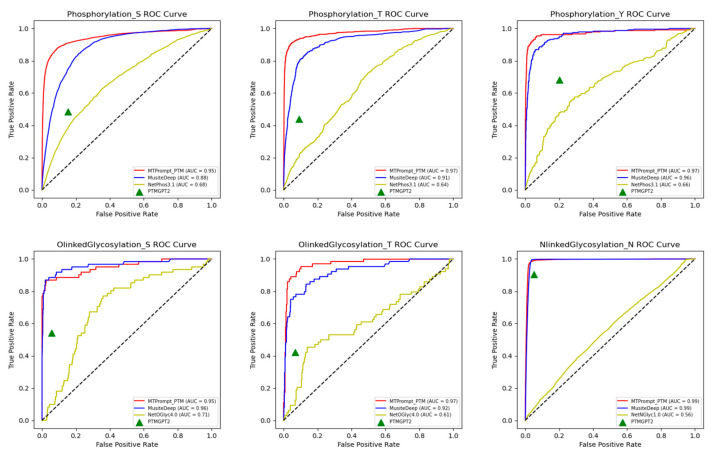
Performance comparison of AUC on 13 PTM types.

**Figure 5 biomolecules-15-00843-f005:**
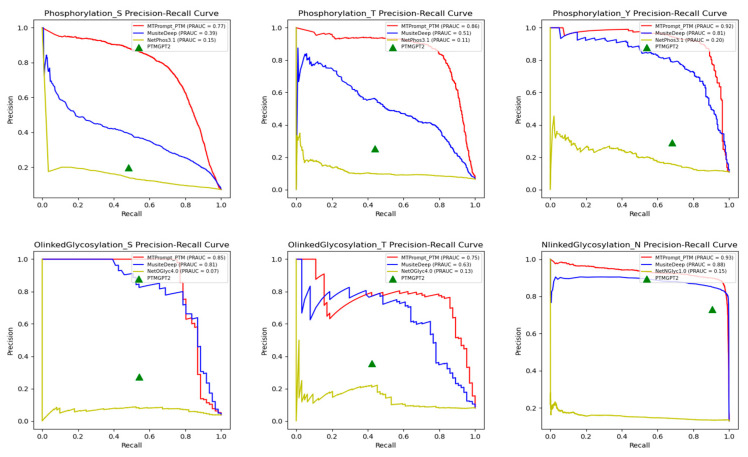
Performance comparison of PRAUC on 13 PTM types.

**Figure 6 biomolecules-15-00843-f006:**
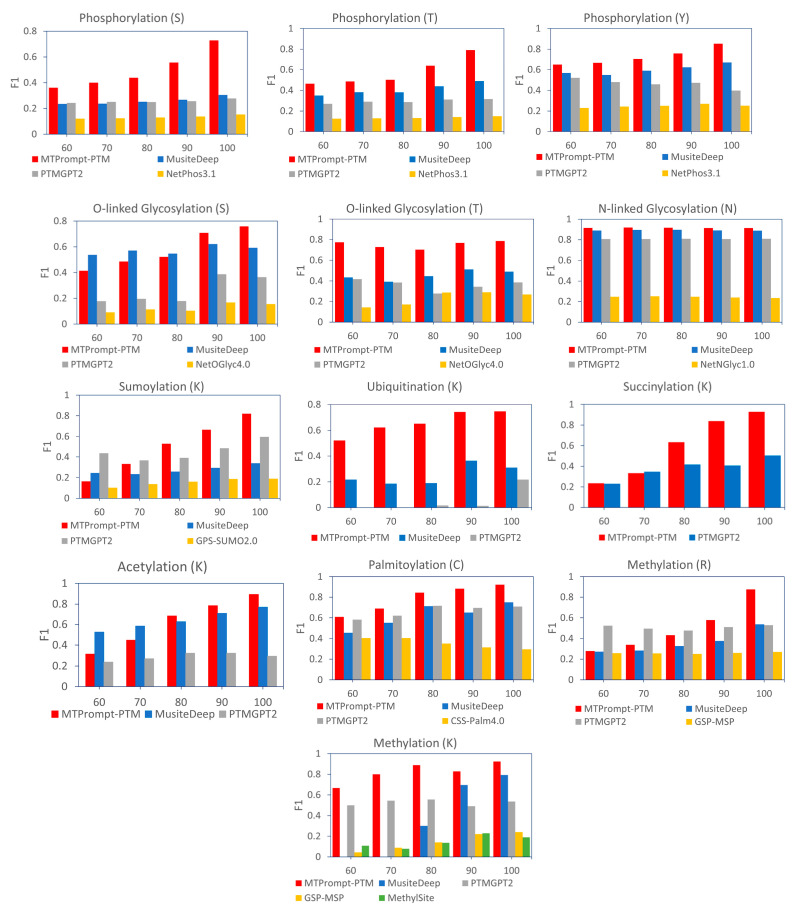
Performance on test subsets with different levels of sequence similarity to the training data, evaluated by F1. The protein sequences of testing data that had no more than 60%, 70%, 80%, 90%, and 100% similarity to the training data were generated by CD-HIT-2D.

**Figure 7 biomolecules-15-00843-f007:**
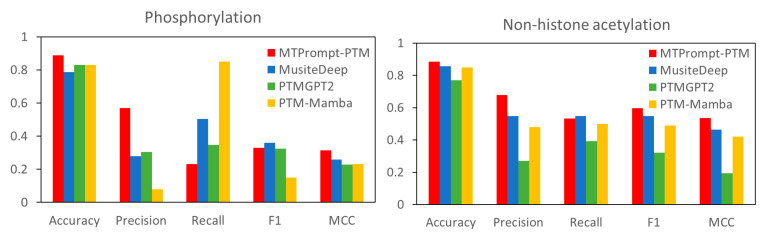
Comparative analysis of PTM prediction tools on independent phosphorylation and acetylation data.

**Figure 8 biomolecules-15-00843-f008:**
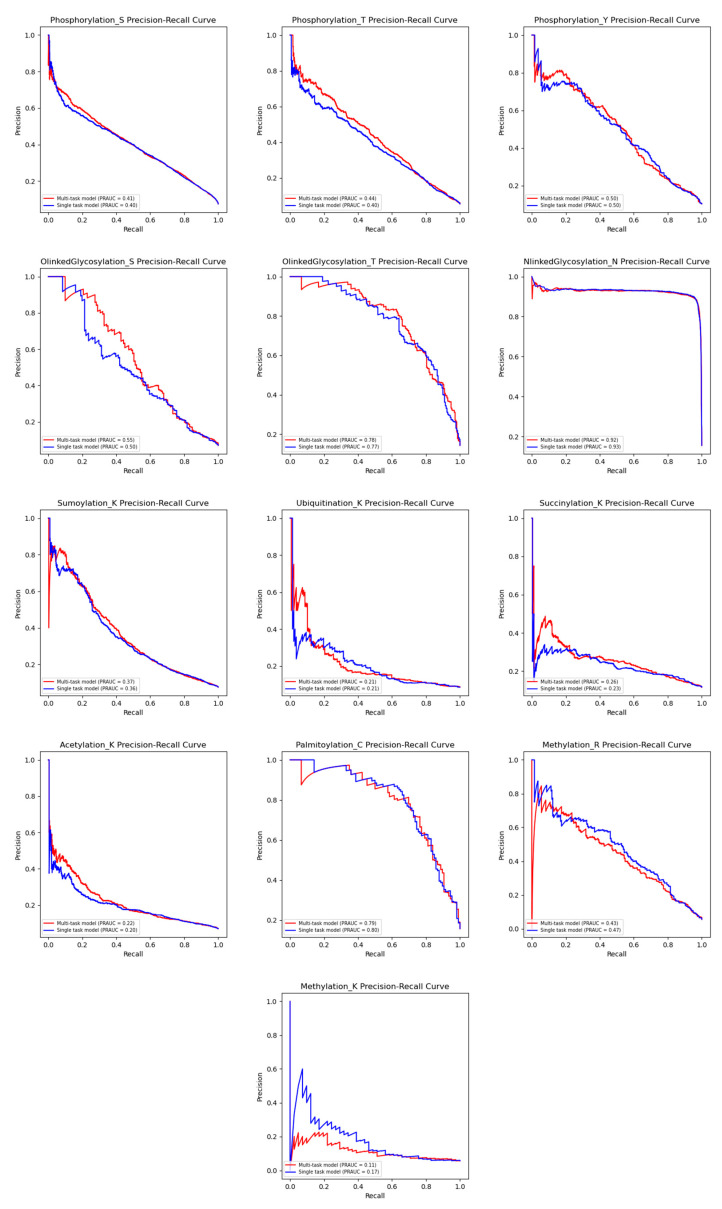
Performance comparison of PRAUC between multi-task and single-task models on 13 PTM types.

**Table 1 biomolecules-15-00843-t001:** Overview of representative PTM prediction tools.

Category	Model	Description	Advantages	Disadvantages
Machine Learning-Based	NetPhos 3.1 NetNGlyc 1.0 GPS-MSP MethylSight Ertelt et al.	Use manually designed features with classical ML models such as ANNs or SVM.	Easy to interpret and efficient for small datasets, producing well-established tools in early PTM prediction research.	Cannot capture long-range dependencies, rely heavily on expert-crafted features, and generalize poorly to unseen data.
Deep Learning-Based	MusiteDeep CapsNet-PTM GPS-SUMO 2.0	Leverage CNNs, CapsuleNets, and other DL architectures to automatically learn features from sequence data.	Automatically learn features from raw data and offer better performance on large-scale datasets.	Rely on local sequence windows, ignore structural information, are usually trained separately for each PTM type, and require large, labeled datasets.
Protein Language Model-Based	Lmnglypred PTG-PLM O-GlcNAc PTM-GPT2 PTM-Mamba	Use embeddings from large-scale pre-trained PLMs or fine-tune PLMs for PTM prediction.	Capture long-range sequence dependencies, benefit from massive pre-training, support transfer learning and generalization.	Lack of direct structural context and rarely leverage effective joint learning across multiple PTM types.

**Table 2 biomolecules-15-00843-t002:** UniProt annotations and number of downloaded protein sequences for different PTM types.

PTM Type	PTM Annotation in UniProt	Number of Protein Sequences
Phosphorylation (S)	Phosphoserine; Diphosphoserine; O-(2-cholinephosphoryl)serine; (Microbial infection) Phosphoserine; O-(pantetheine4′phosphoryl)serine; (Microbial infection) O-(2-cholinephosphoryl) serine	12,230
Phosphorylation (T)	(Microbial infection) Phosphothreonine; Phosphothreonine	8551
Phosphorylation (Y)	Phosphotyrosine	3782
Ubiquitination (K)	(Microbial infection) Glycyl lysine isopeptide (Lys-Gly) (interchain with G-Cter in ubiquitin); Glycyl lysine isopeptide (Lys-Gly) (interchain with G-Cter in ubiquitin and interchain with MARCHF2); Glycyl lysine isopeptide (Lys-Gly) (interchain with G-Cter in ubiquitin)	1225
N-Linked Glycosylation (N)	N-linked (GlcNAc…) (paucimannose) asparagine; N-linked (GlcNAc…) (keratan sulfate) asparagine; N-linked (GlcNAc…) (complex) asparagine; N-linked (GlcNAc) asparagine; N-linked (Glc…) asparagine; N-linked (GlcNAc…) (hybrid) asparagine; N-linked (GalNAc…) asparagine; N-linked (GlcNAc…) (polylactosaminoglycan) asparagine; N-linked (GlcNAc…) asparagine; N-linked (Hex) asparagine; N-linked (HexNAc…) asparagine; N-linked (GlcNAc…) (high mannose) asparagine	12,285
O-Linked Glycosylation (S)	O-linked (Xyl…) (dermatan sulfate) serine; O-linked (Fuc…) serine; O-linked (Xyl…) (heparan sulfate) serine; O-linked (HexNAc…) serine; O-linked (Fuc) serine; O-linked (GalNAc…) serine; O-linked (Xyl…) serine; O-linked (Hex…) serine; O-linked (GlcA) serine; O-linked (GlcNAc) serine; O-linked (GalNAc) serine; O-linked (Man…) serine; O-linked (Xyl…) (glycosaminoglycan) serine; O-linked (Hex) serine; O-linked (GlcNAc…) serine; O-linked (Glc…) serine; O-linked (Xyl…) (chondroitin sulfate) serine; O-linked (Man) serine	942
O-Linked Glycosylation (T)	O-linked (GlcNAc…) threonine; O-linked (Xyl…) (keratan sulfate) threonine; O-linked (Hex) threonine; O-linked (GalNAc) threonine; O-linked (GalNAc…) threonine; O-linked (GlcNAc) threonine; (Microbial infection) O-linked (Glc) threonine; O-linked (Fuc) threonine; O-linked (HexNAc) threonine; O-linked (Man6P…) threonine; O-linked (Man…) threonine; O-linked (Fuc…) threonine; O-linked (HexNAc…) threonine; O-linked (Hex…) threonine; O-linked (Man) threonine	694
Acetylation (K)	N6-acetyllysine; N6-acetyl-N6-methyllysine; (Microbial infection) N6-acetyllysine	6009
Palmitoylation (C)	N-palmitoyl cysteine; S-palmitoyl cysteine	1531
Methylation (R)	Asymmetric dimethylarginine; N5-[4-(S-L-cysteinyl)-5-methyl-1H-imidazol-2-yl]-L-ornithine (Arg-Cys) (interchain with C-151 in KEAP1); Symmetric dimethylarginine; Dimethylated arginine; Omega-N-methylated arginine; Omega-N-methylarginine	1680
Methylation (K)	N6-acetyl-N6-methyllysine; N6-methyllysine; N6,N6,N6-trimethyllysine; N6-methylated lysine; N6,N6-dimethyllysine	578
SUMOylation (K)	Glycyl lysine isopeptide (Lys-Gly) (interchain with G-Cter in SUMO; Glycyl lysine isopeptide (Lys-Gly) (interchain with G-Cter in SUMO1, SUMO2 and SUMO3); Glycyl lysine isopeptide (Lys-Gly) (interchain with G-Cter in /SUMO5); Glycyl lysine isopeptide (Lys-Gly) (interchain with G-Cter in SUMO); Glycyl lysine isopeptide (Lys-Gly) (interchain with G-Cter in SUMO3); Glycyl lysine isopeptide (Lys-Gly) (interchain with G-Cter in SUMO1); Glycyl lysine isopeptide (Lys-Gly) (interchain with G-Cter in SUMO2 and SUMO3); Glycyl lysine isopeptide (Lys-Gly) (interchain with G-Cter in SUMO1 and SUMO2); Glycyl lysine isopeptide (Lys-Gly) (interchain with G-Cter in SUMO1P1/SUMO5); Glycyl lysine isopeptide (Lys-Gly) (interchain with G-Cter in SUMO2)	3724
Succinylation (K)	N6-succinyllysine	2069

**Table 3 biomolecules-15-00843-t003:** Performance comparison between different methods.

PTM Type	Method	Accuracy	F1	MCC	Precision	Recall
Phosphorylation (S)	MTPrompt-PTM	**0.964**	**0.736**	**0.718**	**0.772**	0.704
	MusiteDeep	0.706	0.311	0.328	0.187	**0.917**
	PTMGPT2	0.821	0.281	0.225	0.198	0.483
	NetPhos3.1	0.289	0.155	0.09	0.085	0.905
Phosphorylation (T)	MTPrompt-PTM	**0.975**	**0.811**	**0.798**	**0.787**	**0.836**
	MusiteDeep	0.903	0.512	0.505	0.378	0.793
	PTMGPT2	0.88	0.32	0.272	0.252	0.439
	NetPhos3.1	0.429	0.153	0.103	0.085	0.802
Phosphorylation (Y)	MTPrompt-PTM	**0.973**	**0.873**	**0.858**	**0.877**	**0.87**
	MusiteDeep	0.921	0.705	0.678	0.593	0.87
	PTMGPT2	0.785	0.407	0.342	0.29	0.681
	NetPhos3.1	0.613	0.262	0.154	0.165	0.634
O-Linked Glycosylation (S)	MTPrompt-PTM	**0.983**	**0.774**	**0.765**	**0.762**	0.787
	MusiteDeep	0.956	0.6	0.616	0.454	**0.885**
	PTMGPT2	0.929	0.363	0.351	0.273	0.541
	NetOGlyc4.0	0.718	0.151	0.163	0.085	0.672
O-Linked Glycosylation (T)	MTPrompt-PTM	**0.962**	**0.786**	**0.769**	**0.724**	**0.859**
	MusiteDeep	0.87	0.49	0.47	0.357	0.781
	PTMGPT2	0.892	0.386	0.329	0.355	0.422
	NetOGlyc4.0	0.801	0.267	0.196	0.19	0.453
N-Linked Glycosylation (N)	MTPrompt-PTM	**0.977**	**0.915**	**0.903**	**0.889**	0.944
	MusiteDeep	0.967	0.887	0.875	0.802	**0.992**
	PTMGPT2	0.944	0.808	0.782	0.73	0.905
	NetNGlyc1.0	0.294	0.233	0.033	0.136	0.825
SUMOylation (K)	MTPrompt-PTM	**0.97**	**0.838**	**0.823**	**0.878**	0.802
	MusiteDeep	0.901	0.332	0.299	0.472	0.256
	PTMGPT2	0.92	0.606	0.563	0.572	0.644
	GPS-SUMO2.0	0.187	0.187	0.081	0.103	**0.978**
Ubiquitination (K)	MTPrompt-PTM	**0.956**	**0.782**	**0.764**	**0.879**	0.704
	MusiteDeep	0.681	0.367	0.318	0.236	**0.831**
	PTMGPT2	0.634	0.259	0.14	0.167	0.575
Succinylation (K)	MTPrompt-PTM	**0.983**	**0.933**	**0.923**	**0.944**	**0.922**
	PTMGPT2	0.834	0.519	0.452	0.408	0.715
Acetylation (K)	MTPrompt-PTM	**0.981**	**0.899**	**0.889**	**0.924**	0.877
	MusiteDeep	0.948	0.778	0.762	0.671	**0.926**
	PTMGPT2	0.724	0.286	0.199	0.192	0.562
Palmitoylation (C)	MTPrompt-PTM	**0.974**	**0.926**	**0.91**	**0.943**	0.909
	MusiteDeep	0.916	0.759	0.708	0.774	0.745
	PTMGPT2	0.883	0.695	0.626	0.651	0.745
	CSS-Palm4.0	0.183	0.3	-0.03	0.177	**0.982**
Methylation (R)	MTPrompt-PTM	**0.989**	**0.892**	**0.889**	**0.967**	0.829
	MusiteDeep	0.915	0.565	0.59	0.399	**0.967**
	PTMGPT2	0.921	0.54	0.54	0.403	0.819
	GSP-MSP	0.734	0.274	0.304	0.162	0.881
Methylation (K)	MTPrompt-PTM	**0.976**	**0.883**	**0.87**	**0.901**	0.867
	MusiteDeep	0.952	0.791	0.768	0.728	0.867
	PTMGPT2	0.869	0.529	0.477	0.427	0.695
	GSP-MSP	0.337	0.234	0.162	0.133	**0.962**
	MethylSight	0.384	0.19	0.022	0.11	**0.686**

Note: Numbers in bold represent the highest values achieved for each metric (Accuracy, F1, MCC, Precision, Recall) within a given PTM type.

**Table 4 biomolecules-15-00843-t004:** Performance comparison of PTM prediction models across different kinase families.

Kinase Type	Number of Kinase Sites in Training Set	Method	Number of Kinase Sites in Testing Set	Number of Predicted Kinase Sites in Testing Set	Accuracy
AGC	879	MTPrompt-PTM	17	15	**0.882**
		MusiteDeep	17	14	0.824
		PTMGPT2	17	7	0.412
		NetPhos3.1	17	12	0.706
CAMK	392	MTPrompt-PTM	2	1	**0.5**
		MusiteDeep	2	2	1
		PTMGPT2	2	1	**0.5**
		NetPhos3.1	2	2	1
CK1	48	MTPrompt-PTM	1	1	1
		MusiteDeep	1	1	1
		PTMGPT2	1	0	0
		NetPhos3.1	1	1	1
CMGC	1739	MTPrompt-PTM	47	43	**0.9** **15**
		MusiteDeep	47	42	0.894
		PTMGPT2	47	21	0.447
		NetPhos3.1	47	40	0.851
Other	415	MTPrompt-PTM	9	1	0.111
		MusiteDeep	9	3	**0.333**
		PTMGPT2	9	1	0.111
		NetPhos3.1	9	2	0.222
STE	174	MTPrompt-PTM	2	2	**1**
		MusiteDeep	2	2	**1**
		PTMGPT2	2	2	**1**
		NetPhos3.1	2	2	**1**
TK	753	MTPrompt-PTM	4	4	**1**
		MusiteDeep	4	4	**1**
		PTMGPT2	4	2	0.5
		NetPhos3.1	4	2	0.5

Note: Numbers in bold represent the highest values achieved for accuracy within a given kinase type.

**Table 5 biomolecules-15-00843-t005:** Performance comparison of F1 and MCC on MTPrompt-PTM and separately trained model.

	F1/MCC	
PTM Type (Residue)	Multi-Task Model	Single-Task Model
Phosphorylation (S)	0.428/**0.384**	**0.429**/0.383
Phosphorylation (T)	**0.461/0.432**	0.439/0.406
Phosphorylation (Y)	**0.503/0.459**	0.498/0.448
N-Linked Glycosylation (N)	0.918/0.902	**0.922**/**0.907**
O-Linked Glycosylation (S)	**0.524/0.5**	0.487/0.447
O-Linked Glycosylation (T)	**0.716/0.685**	0.670/0.634
Palmitoylation (C)	**0.74/0.697**	0.730/0.685
Acetylation (K)	**0.214/0.206**	0.189/0.180
Ubiquitination (K)	**0.081/0.129**	0.051/0.074
Succinylation (K)	**0.208/0.176**	0.144/0.109
SUMOylation (K)	0.352/**0.342**	**0.361**/0.332
Methylation (K)	0/0.142	**0.089**/**0.143**
Methylation (R)	0.431/0.414	**0.470/0.454**

Note: Numbers in bold represent the highest values achieved for F1 and MCC within a given PTM type.

**Table 6 biomolecules-15-00843-t006:** Performance comparison of F1 and MCC on MTPrompt-PTM and multi-task model without knowledge distillation.

	F1/MCC	
PTM Type (Residue)	Multi-Task Model with Knowledge Distillation	Multi-Task Model Without Knowledge Distillation
Phosphorylation (S)	**0.428/0.384**	0.341/0.338
Phosphorylation (T)	**0.461/0.432**	0.389/0.381
Phosphorylation (Y)	**0.503/0.459**	0.448/0.424
N-Linked Glycosylation (N)	**0.918/0.902**	0.916/0.901
O-Linked Glycosylation (S)	**0.524/0.5**	0.45/0.446
O-Linked Glycosylation (T)	**0.716/0.685**	0.667/0.634
Palmitoylation (C)	**0.74/0.697**	0.719/0.678
Acetylation (K)	**0.214/0.206**	0.160/0.164
Ubiquitination (K)	**0.081/0.129**	0.081/0.129
Succinylation (K)	**0.208/0.176**	0.044/0.062
SUMOylation (K)	**0.352/0.342**	0.253/0.301
Methylation (K)	**0/0.142**	0/0.142
Methylation (R)	**0.431**/0.414	0.411/**0.415**

Note: Numbers in bold represent the highest values achieved for F1 and MCC within a given PTM type.

**Table 7 biomolecules-15-00843-t007:** Performance comparison of AUROC and AUPRC of MTPrompt-PTM and multi-task model without knowledge distillation.

	AUROC/AUPRC	
PTM Type	Multi-Task Model with Knowledge Distillation	Multi-Task Model Without Knowledge Distillation
Phosphorylation (S)	**0.866**/0.409	0.866/**0.411**
Phosphorylation (T)	0.878/**0.436**	**0.879**/0.429
Phosphorylation (Y)	0.844/**0.504**	**0.845**/0.496
N-Linked Glycosylation (N)	0.990/0.924	**0.991**/**0.927**
O-Linked Glycosylation (S)	**0.870/0.552**	0.864/0.518
O-Linked Glycosylation (T)	**0.933/0.784**	0.933/0.761
Palmitoylation (C)	**0.929**/0.791	0.925/**0.792**
Acetylation (K)	0.739/**0.220**	**0.742**/0.212
Ubiquitination (K)	**0.669/0.215**	0.669/0.215
Succinylation (K)	0.720/0.260	**0.722/0.268**
SUMOylation (K)	**0.798/0.369**	0.797/0.359
Methylation (K)	0.663/0.122	**0.669/0.130**
Methylation (R)	0.893/0.438	**0.910/0.450**

Note: Numbers in bold represent the highest values achieved for AUROC and AUPRC within a given PTM type.

**Table 8 biomolecules-15-00843-t008:** Performance comparison of F1 and MCC of MTPrompt-PTM and multi-task model with fine-tuning in the last two layers of S-PLM.

	F1/MCC	
PTM Type	Multi-Task Model with Prompt Tuning	Multi-Task Model with Fine-Tuning in Last Two Layers of S-PLM v2
Phosphorylation (S)	**0.428/0.384**	0.355/0.340
Phosphorylation (T)	**0.461/0.432**	0.403/0.392
Phosphorylation (Y)	**0.503/0.459**	0.450/0.427
N-Linked Glycosylation (N)	**0.918/0.902**	0.916/0.900
O-Linked Glycosylation (S)	**0.524/0.5**	0.49/0.481
O-Linked Glycosylation (T)	**0.716/0.685**	0.548/0.554
Palmitoylation (C)	**0.74/0.697**	0.695/0.651
Acetylation (K)	**0.214/0.206**	0.214/0.206
Ubiquitination (K)	0.081/0.129	**0.082/0.156**
Succinylation (K)	**0.208/0.176**	**0.11/0.124**
SUMOylation (K)	**0.352/0.342**	**0.268/0.304**
Methylation (K)	0/0.142	0.048/0.152
Methylation (R)	0.431/0.414	0.435/0.435

Note: Numbers in bold represent the highest values achieved for F1 and MCC within a given PTM type.

**Table 9 biomolecules-15-00843-t009:** Performance comparison of AUROC and AUPRC of MTPrompt-PTM and multi-task model with fine-tuning in the last two layers of S-PLM.

	AUROC/AUPRC	
PTM Type	Multi-Task Model with Prompt Tuning	Multi-Task Model with Fine-Tuning in Last Two Layers of S-PLM v2
Phosphorylation (S)	**0.866/0.409**	0.865/0.409
Phosphorylation (T)	0.878/**0.436**	**0.882**/0.434
Phosphorylation (Y)	**0.844/0.504**	0.837/0.499
N-Linked Glycosylation (N)	0.990/0.924	**0.991**/**0.930**
O-Linked Glycosylation (S)	**0.870/0.552**	0.845/0.525
O-Linked Glycosylation (T)	**0.933/0.784**	0.922/0.757
Palmitoylation (C)	**0.929**/0.791	0.927/**0.795**
Acetylation (K)	**0.739/0.220**	0.739/0.220
Ubiquitination (K)	0.669/**0.215**	**0.672**/0.207
Succinylation (K)	**0.720/0.260**	0.717/0.259
SUMOylation (K)	**0.798/0.369**	0.792/0.354
Methylation (K)	0.663/0.122	**0.704/0.289**
Methylation (R)	0.893/0.438	**0.903/0.461**

Note: Numbers in bold represent the highest values achieved for AUROC and PRAUC within a given PTM type.

## Data Availability

The source code, data, and trained model are available at GitHub (https://github.com/hanye311/MTPrompt-PTM/) (accessed on 6 June 2025).

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
