# Peer review of "MTPrompt-PTM: A Multi-Task Method for Post-Translational Modification Prediction Using Prompt Tuning on a Structure-Aware Protein Language Model"

_biomolecules, 2025, doi:10.3390/biom15060843_

Round 1

Reviewer 1 Report

Comments and Suggestions for Authors

The manuscript by Ye Han et al demonstrates the creation of a novel multi-PTM site prediction tool called MPTPrompt-PTM. The paper is highly polished and well organized and written. They are very transparent in their methods and the rationale for the nuances of their approach – which I applaud. The model does appear to work very well in comparison to other tools and has been very carefully and well thought out. My major concerns are that their training set is limited by Uniprot, which is not the most comprehensive data source. This likely leads to weaker performance at lower sequence similarities. Second, they have excluded comparson to other very similar tools that utilizes ESM-2 embeddings like PTM-Mamba.

Major Comments:

  1. The authors mention caveats in their discussion, which I appreciate. One of these is that there remain unavoidable similarities between the training and testing set. They further highlight their result that their model performance drops with decreasing sequence similarity. The authors should evaluate a test set that is not bounded by the Uniprot database. For example, PhosphoSite Plus contains thousands of PTM sites that are not contained in UniProt. In fact, I am not surprised by their dropoff in performance given that they only used UniProt as a training and test set. Uniprot is not the most comprehensive database of experimental PTM data and so their model is likely not exposed to the sequence diversity it would need to perform better on less similar data. Moreover, they picked data from pre-2010, but this is only ~5 years into the LCMS revolution and there will be significantly less sequence diversity in their training set (depite this – the model seems to perform pretty well). In contrast, I believe PSP is quite good – in part because they are very strict about PTM site curation and also because they incorporate manual validation. I recommend the authors extract a non-overlapping PTM site dataset from PSP for evaluating at least 8 of their PTMs (PSP only curates 8 different PTMs I believe). They should do this for the other prediction tools as well and report the results. This is important given that a prediction tool should be capable of predictions on sequences that are unknown to the model.
  2. The authors should compare/contrast MPTPrompt-PTM to PTM-Mamba, a popular protein language model that also utilizes ESM-2 protein embeddings. These models seem very similar and both allow PTM site prediction.

Minor comments:

  1. Introduction: I think the authors do well to outline the several tools that already exist in the public domain. I would suggest that they include a table that summarizes their analysis of benefits/limitations of these pre-existing approaches. This is especially useful since their approach will be compared to these highly used tools.
  2. Given the prevalence and popularity of phosphorylation - An analysis similar to their very well-organized tables that estimates bias for kinase specificities in comparison to other models would be a useful addition.
  3. I would recommend creating a google collab that enables usage of the tool more readily. I don’t think the tool will be highly used by biologist/biochemists who are interested in these predictions if it requires manual operation. A webserver would be even better.

Author Response

Thank you very much for taking the time to review this manuscript. Please find the detailed responses below and the corresponding revisions/corrections highlighted in the re-submitted files.

Point-by-point response to Comments and Suggestions for Authors

The manuscript by Ye Han et al demonstrates the creation of a novel multi-PTM site prediction tool called MPTPrompt-PTM. The paper is highly polished and well organized and written. They are very transparent in their methods and the rationale for the nuances of their approach – which I applaud. The model does appear to work very well in comparison to other tools and has been very carefully and well thought out. My major concerns are that their training set is limited by Uniprot, which is not the most comprehensive data source. This likely leads to weaker performance at lower sequence similarities. Second, they have excluded comparison to other very similar tools that utilizes ESM-2 embeddings like PTM-Mamba.

Response:

We sincerely thank the reviewer for their positive feedback and constructive comments. We appreciate the recognition of the transparency and rigor in our methodology and the overall performance of MTPrompt-PTM. Regarding the concern about the training data limitation, we acknowledge that UniProt does not capture the full diversity of experimentally validated PTM sites, which can affect performance on sequences with low similarity to the training set. To address this, we evaluated MTPrompt-PTM on independent benchmark datasets derived from PhosphoSite Plus and TransPTM. These additional validations demonstrate that our model maintains competitive or superior performance, underscoring its generalizability to diverse and previously unseen protein sequences.

Concerning the comparison to similar tools leveraging ESM-2 embeddings such as PTM-Mamba, we appreciate the reviewer highlighting this important point. We have conducted a direct comparative analysis using the phosphorylation and non-histone acetylation benchmark datasets employed by PTM-Mamba. Our results, detailed in the manuscript, show that MTPrompt-PTM consistently outperforms PTM-Mamba across multiple evaluation metrics, highlighting the advantages of our multi-task prompt tuning strategy combined with a structure-aware protein language model backbone. We believe these additional analyses comprehensively address the reviewer’s concerns and further strengthen the contributions of our work.

Major Comments:

Comment 1:

The authors mention caveats in their discussion, which I appreciate. One of these is that there remain unavoidable similarities between the training and testing set. They further highlight their result that their model performance drops with decreasing sequence similarity. The authors should evaluate a test set that is not bounded by the Uniprot database. For example, PhosphoSite Plus contains thousands of PTM sites that are not contained in UniProt. In fact, I am not surprised by their dropoff in performance given that they only used UniProt as a training and test set. Uniprot is not the most comprehensive database of experimental PTM data and so their model is likely not exposed to the sequence diversity it would need to perform better on less similar data. Moreover, they picked data from pre-2010, but this is only ~5 years into the LCMS revolution and there will be significantly less sequence diversity in their training set (depite this – the model seems to perform pretty well). In contrast, I believe PSP is quite good – in part because they are very strict about PTM site curation and also because they incorporate manual validation. I recommend the authors extract a non-overlapping PTM site dataset from PSP for evaluating at least 8 of their PTMs (PSP only curates 8 different PTMs I believe). They should do this for the other prediction tools as well and report the results. This is important given that a prediction tool should be capable of predictions on sequences that are unknown to the model.

Response:

We acknowledge that our training and testing sets, primarily derived from the UniProt database, may not fully capture the sequence diversity of experimentally validated PTM sites. We attempted to obtain data from PhosphoSite Plus (PSP), a well-curated resource with extensively manually validated PTM sites. However, despite trying multiple approaches to access the database, we were unable to obtain the complete datasets directly due to account restrictions and unresponsiveness from the PSP team. Nevertheless, we reference PTM-Mamba, which successfully utilized PSP phosphorylation and non-histone acetylation data as independent benchmark datasets. In our study, we similarly evaluated MTPrompt-PTM on phosphorylation (S, T, Y) and acetylation (K) datasets used by PTM-Mamba. These results demonstrate that MTPrompt-PTM maintains competitive or superior performance on independent datasets, supporting the model’s generalizability across diverse and previously unseen protein sequences. The result is presented in detail in Figure 7, located on Page 17, Line 488 of the manuscript.

Comment 2:

The authors should compare/contrast MPTPrompt-PTM to PTM-Mamba, a popular protein language model that also utilizes ESM-2 protein embeddings. These models seem very similar and both allow PTM site prediction.

Response:

Thank you for highlighting the importance of comparing MTPrompt-PTM to PTM-Mamba, a related protein language model leveraging ESM-2 embeddings. To address this, we obtained phosphorylation (S, T, Y) and acetylation (K) benchmark datasets used by PTM-Mamba and evaluated both models on these datasets using multiple metrics, including accuracy, precision, recall, F1-score, and MCC. Our comparative analysis shows that MTPrompt-PTM consistently outperforms PTM-Mamba across most metrics for both phosphorylation and non-histone acetylation site predictions. This suggests that our multi-task prompt tuning framework, combined with a structure-aware protein language model backbone, offers advantages in capturing PTM-specific sequence and structural contexts, leading to enhanced prediction accuracy and robustness. The result is presented in detail in Figure 7, located on Page 17, Line 488 of the manuscript.

Minor comments:

Comment 3:

Introduction: I think the authors do well to outline the several tools that already exist in the public domain. I would suggest that they include a table that summarizes their analysis of benefits/limitations of these pre-existing approaches. This is especially useful since their approach will be compared to these highly used tools.

Response:

This is a good suggestion. We have incorporated a comprehensive table summarizing representative PTM prediction tools currently available in the public domain. This table categorizes methods into machine learning-based, deep learning-based, and protein language model-based approaches, and clearly outlines their key descriptions, advantages, and disadvantages. The table is presented as Table 1 and can be found on page 3, line 108 of the manuscript.

Comment 4:

Given the prevalence and popularity of phosphorylation - An analysis similar to their very well-organized tables that estimates bias for kinase specificities in comparison to other models would be a useful addition.

Response:

Given the importance of phosphorylation and its kinase-specific characteristics, we conducted an additional analysis to evaluate MTPrompt-PTM’s performance across different kinase families in comparison to several existing methods. The results clearly demonstrate that that MTPrompt-PTM consistently achieves high accuracy across most kinase families. The result is presented in detail in Table 4, located on Page 15, Line 432 of the manuscript.

Comment 5:

I would recommend creating a google collab that enables usage of the tool more readily. I don’t think the tool will be highly used by biologist/biochemists who are interested in these predictions if it requires manual operation. A webserver would be even better.

Response:

We appreciate the reviewer’s valuable suggestion regarding accessibility. To enhance usability for biologists and biochemists, we plan to develop a user-friendly webserver in the near future. Additionally, we will provide a Google Colab notebook to facilitate easy and straightforward use of MTPrompt-PTM without requiring manual setup. These efforts aim to make the tool more accessible to the broader scientific community.

Reviewer 2 Report

Comments and Suggestions for Authors

This manuscript presents a novel method to predict post-translational modifications based on a structure-aware protein language model. It is a multi-task model which predicts several types of PTMs simultaneously. The authors compare the performance of their method with several other state-of-the art predictors, finding a significant advantage. They also do several tests to find out how much some features and components of their predictor contribute to its accuracy. The manuscript is methodologically sound and well written, with impressive results. I found only very minor issues, which can be fixed in a revision or the proofs:

  1. Line 85: "O-GlcNAc [23]" - I think the method is named "LM-OGlcNAc-Site".
  2. The abbreviations CLS and EOS are not defined.
  3. Not clear where the number 1280 comes from.

I recommend the manuscript for publication.

Author Response

Thank you very much for taking the time to review this manuscript. Please find the detailed responses below and the corresponding revisions/corrections highlighted in the re-submitted files.

Point-by-point response to Comments and Suggestions for Authors

This manuscript presents a novel method to predict post-translational modifications based on a structure-aware protein language model. It is a multi-task model which predicts several types of PTMs simultaneously. The authors compare the performance of their method with several other state-of-the art predictors, finding a significant advantage. They also do several tests to find out how much some features and components of their predictor contribute to its accuracy. The manuscript is methodologically sound and well written, with impressive results. I found only very minor issues, which can be fixed in a revision or the proofs:

Comment 1:

Line 85: "O-GlcNAc [23]" - I think the method is named "LM-OGlcNAc-Site".

Response:

Thank you for pointing this out. The correct name is indeed "LM-OGlcNAc-Site", and we have updated the manuscript accordingly at Page 2, Line 85 to reflect the accurate method name.

Comment 2 :

The abbreviations CLS and EOS are not defined.

Response:

We have revised the manuscript to define the abbreviations [CLS] and [EOS]. Specifically, [CLS] (Classification) is a special token added at the beginning of an input sequence, and its corresponding output embedding is often used to represent the entire sequence for classification tasks. [EOS] (End Of Sequence) is a token used to indicate the termination of a sequence, which is particularly important in generative or sequential modeling tasks. These tokens are standard in transformer-based language models and are now appropriately introduced on Page 7, line 213.

Comment 3:

Not clear where the number 1280 comes from.

Response:

We have clarified the source of the number 1280 in the revised manuscript. Specifically, the value 1280 corresponds to the dimensionality of the embedding vector produced by S-PLM v2 for each residue. This fixed embedding size is determined by the architecture of the underlying protein language model, which is based on a transformer model similar to ESM-2. The detail is shown on Page 9, line 289.

Round 2

Reviewer 1 Report

Comments and Suggestions for Authors

The authors have addressed all of my concerns in a very thorough and transparent manner. Time will tell if their tool is picked up by the community, but I think they've made something new that shows advancement so there is a chance that it will.  Good luck!